# Research Trends and Methodological Approaches of the Impacts of Windstorms on Forests in Tropical, Subtropical, and Temperate Zones: Where Are We Now and How Should Research Move Forward?

**DOI:** 10.3390/plants9121709

**Published:** 2020-12-04

**Authors:** Jonathan O. Hernandez, Lerma S.J. Maldia, Byung Bae Park

**Affiliations:** 1Department of Environment and Forest Resources, College of Agriculture and Life Sciences, Chungnam National University, Daejeon 34134, Korea; johernandez2@up.edu.ph; 2Department of Forest Biological Sciences, College of Forestry and Natural Resources, University of the Philippines Los Baños, Laguna 4031, Philippines; lsmaldia@up.edu.ph

**Keywords:** experimental forest, hurricane, observational study, PRISMA, systematic review article, tree mortality, tropical cyclones, typhoon, wind throw

## Abstract

Windstorm is one of the destructive natural disturbances, but the scale-link extent to which recurrent windstorms influenced forests ecosystems is poorly understood in a changing climate across regions. We reviewed the synergistic impacts of windstorms on forests and assessed research trends and methodological approaches from peer-reviewed articles published from 2000 to 2020 in tropical (TRF), subtropical (SUF), and temperate (TEF) forests/zones, based on the Preferred Reporting Items for Systematic Reviews and Meta-Analyses (PRISMA) guidelines. Overall, the majority of the reviewed studies were conducted in TRF (i.e., 40%), intermediate in SUF (i.e., 34%), and the lowest in TEF (i.e., 26%). Among the four levels of biological organization, the species-population and community-ecosystem levels had the highest number of study cases, while the molecular-cellular-individual and landscape levels had the lowest study cases in all forest types. Most of the articles reviewed dealt largely on tree mortality/survival and regeneration/succession for TRF, tree mortality/survival and species composition/richness/diversity for SUF, and stem density, gap dynamics, and regeneration/succession for TEF. However, research on the effects of windstorms on mycorrhizal symbioses, population genetics, and physiological adaptation, element fluxes via litterfall, litter decomposition, belowground processes, biological invasion, and tree health are less common in all forest types. Further, most of the studies were conducted in permanent plots but these studies mostly used observational design, while controlled studies are obviously limited. Consequently, more observational and controlled studies are needed on the topic reviewed, particularly studies at the molecular-cellular-individual and landscape levels, to help inform forest management decision-making about developing sustainable and resilient forests amid climate change.

## 1. Introduction

Nearly all forest ecosystems throughout the world are shaped and influenced by natural disturbances, including cyclonic windstorms (hurricanes, cyclones, and tornadoes). Catastrophic storms cause immediate and long-term structural damage to individual trees, including massive defoliation and branch loss, a decline in stem density, basal area, diameter at breast height (DBH), and total height [1,2]. More specifically, the structure of the forest canopy can be molded by frequent wind disturbance, influencing the biophysical environment, tree physiology, atmospheric exchange, and biotic habitat [3,4]. Windstorms also shape the functional composition of forests by creating selection pressures via increased stem mortality, reduced reproduction, facilitated natural regeneration, and dominance of pioneer species characterized by less dense wood and enhanced plant functional traits (e.g., higher specific leaf area, low seed mass) [5,6]. The forest damage often varies in intensity of windstorms resulting in high variation in tree species and ecosystem damage, secondary succession, and trajectory of forest recovery [7,8].

The profound effects of windstorms on forests had long been studied, but most of these, if not limited to a particular area, had been focused primarily on short-term impacts on individual trees, such as defoliation, branch loss, and canopy disturbance [9,10,11]. Later on, much progress has been reported in explaining visible windstorms’ impacts on forest physical environment, tree regeneration dynamics, and forest recovery in temperate and tropical regions [12,13,14], resulting in the evolution of a contemporary view of windstorms as a multi-scale forest ecological disturbance [15,16,17]. Recently, a study of Lin [18] has shown a scale-link perspective of tropical cyclone ecology, in which the effects of cyclones at the community and ecosystem levels are linked to effects at the individual and species levels. However, the synergistic impacts of catastrophic windstorms on forests across levels of biological organizations has remained poorly understood and relatively unpredictable due to the unresolved issues relevant to the influence of climate change on windstorm characteristics. Windstorms have a significant role in characterizing ecosystem stability and dynamics, whose behavior and nature of damage change with climate change, with storms increasing in intensity and size [19,20]. Climate change affects not just the patterns of precipitation and temperature but also the frequency of cyclonic windstorms, which indirectly influences the forest carbon and emissions globally. The impacts are also aggravated by many interacting meteorological (e.g., wind speed, rainfall, and temperature), topographical (e.g., exposure), and biological factors (e.g., trees species and size), causing a complex alteration in forest function and dynamics across landscapes. Moreover, most of the understandings of tropical cyclones on forests were generally biased towards high latitudes regions, such as Canada, the USA, and New Zealand [21,22,23,24]. Some of these cited studies have also mentioned concerns regarding the methodological approaches for acquiring pre- and post-cyclone data across regions. This knowledge gap limits our understanding of the complex effects of windstorms, particularly on forests structure, species composition, carbon uptake and emissions, and forest productivity in the context of meeting emission targets for greenhouse gases and creating sustainable and resilient forests across geographical regions. Thus, further research about the impacts of windstorms on forests and ecosystems post-disturbance monitoring are needed to enhance our understanding of windstorm ecology amid climate change.

Here we define windstorms as the large-scale wind events, including hurricanes, typhoons, and cyclonic storms characterized by strong winds and heavy precipitation, which greatly influence many aspects of a forest from individual tree growth, tree regeneration, and community structure to ecosystem function [18,25]. Large scale analyses have constantly detected significant factors, including tree species, size, density, and shape, wood traits, soil characteristics, weather, topography, and altitude, influencing windstorm risk of trees and stands [26,27].

The purpose of this systematic review is to update the framework of understanding the synergistic impacts of windstorms on forests and assessed research trends and methodological approaches in tropical (TRF), subtropical (SUF), and temperate (TEF) from peer-reviewed articles published from 2000 to 2020. Such time period selection considered the necessary degree of comprehension that is appropriate for the literature search questions that we would like to investigate through a preliminary online literature search. In addition to the identification of knowledge gaps about the reviewed topic, we also structured the present review into the following questions: (1) what is the proportion of studies for tropical versus subtropical and temperate regions?; (2) which effects on forests are mostly examined at each biological organization and forest functional type?; and (3) which methodological approaches were commonly used in determining the effects of windstorms on the forest?. The present work will provide us with an integrative understanding of the impacts of catastrophic windstorms on forests, particularly on structure, species composition, carbon storage and emission, and dynamics of tropical, subtropical, and temperate forests across levels of biological organization (i.e., molecular to landscape levels). Thus, the synthesis will help us to understand how future changes in windstorm frequency and size affect the forest ecosystems, and inform our decision-making for improved forest management.

## 2. Results

### 2.1. Research Topics Across Levels of Biological Organization

Out of 11,017 articles from Science Direct and Scopus online databases, a total of 161 peer-reviewed articles that were published between 2000 and 2020 met the inclusion criteria (see Section 4.2). The majority of the articles were conducted in TRF (40%), followed by SUF (34%) and TEF (26%) (Figure 1). Among the four levels of biological organization, the species-population and ecosystem-community levels had the highest number of research topics in all forest types/zones. In TRF, the most studied effects of a windstorm on the forest are tree mortality/survival (24.6%) for species-population level and regeneration/succession (15.5%) for the community-ecosystem level. In SUF, articles about tree mortality/survival had the highest percentage of study cases (i.e., 22.2%) for species-population level followed by species composition/richness/diversity with 16.5% cases for community-ecosystem level. The stem density, gap dynamics, and regeneration/succession were the most studied effects of windstorms, all with 19.0% study cases, in TEF (Figure 1). Contrarily, there were only five to six research topics (5–10% of the total topics) that have been investigated at the molecular-cellular-individual and landscape levels in all forest types.

Moreover, some important research topics at the higher level of the biological organization had low percentage (215%) of published study cases in all forest types/zones. These topics include tree health, phenology, seed traits/seed bank dynamics, population dynamics/demography, fine root, soil microbial communities, foliar nitrogen (N) concentration, litterfall and litter decomposition, invasion dynamics, carbon (C) and nutrient dynamics (particularly, non-photosynthetic vegetation (NPV)), and CO_2_ fluxes/sequestration/storage (particularly in TRF and SUF). Research areas including basal area (BA)/diameter at breast height (DBH)/height, biomass and productivity, and Normalized Difference Vegetation Index (NDVI) were generally either second or third in rank in terms of a number of study cases in all forest types.

### 2.2. Study Approaches to Determine Windstorm Effects Across Forest Types

The reviewed studies were conducted in either permanent or temporary plots established across the three geographical zones (Figure 2). Most of the permanent/experimental plots were established in experimental forests in the national forest reserves or protected areas. Generally, there was no big difference in the count of studies between the two plot types in all forests types/zones. In terms of most used permanent plots, a total of 17 studies were conducted in Luquillo Experimental Forest in Puerto Rico for TRF, six studies were conducted in Fushan Experimental Forest in Taiwan for SUF, and three studies were carried out in Tomakomai Experimental Forest in Japan for TEF. The other experimental forests had only 12 study cases.

Not surprisingly, a two-fold higher number of studies in TRF than in the SUF or TEF was observed in this review. However, we found that some of the windstorm-prone countries in the tropics, such as Vietnam, Philippines, China, Mexico, and Australia have no to very few study cases about windstorms effects on forest ecosystems, particularly those conducted in experimental/permanent sites (Figure 2).

Majority of the most-studied research topics used observational approach in determining the effects of windstorms in all forest types (Figure 3a). In terms of length of study duration, the median values are ranging from 2–6 years for TRF, 4-14 years for SUF, and 5–7 years for TEF (Figure 3a). There were two to eight articles which were conducted in as long as 30–33, 77–78, and 25–38 years at TRF, SUF, and TEF, respectively. Several studies were also conducted in less than one year, that is, 1 to 9 months study duration in TRF and 1 to 2 months in SUF and TEF. Moreover, we found that most of these studies were simultaneously conducted across the three geographical zones, starting from 1997–2005. However, there was a conspicuous gap in research before and after that period (Figure 3b) in all forest types/zones.

## 3. Discussion

### 3.1. Synergistic Effects of Catastrophic Windstorms on Forest Across Levels of Biological Organization

#### 3.1.1. Individual-Level Effects

Generally, the effect of windstorms on forests may start from the effects on individual trees through defoliation, snapping, uprooting, and crown damage (Figure 4). These immediate effects can vary depending on the resistance properties and wood anatomical and mechanical traits of trees [28,29], stem biometrics [30], level of exposure, and tree health [18]. The wood density, for instance, has a strong correlation with tropical cyclone damage, such as defoliation, limb loss, snapping, and uprooting in tropical rainforests in Australia [31]. Wood density is related to the mechanical strength of a stem or wood stiffness; thus, tree species with higher wood density (e.g., >1.39 g cm^−3^) have shown lower cyclone damage in several countries such as Puerto Rico, Philippines, Taiwan, and Australia [31,32,33,34]. However, there was also evidence that trees with higher wood density (e.g., those with higher stem biomass costs) are less prone to snapping but more susceptible to uprooting, making wood density as an important factor determining windthrow resistance in both tropical and temperate forests [35,36]. This wood trait was also found to negatively correlated to the proportion of defoliated trees after Hurricane Jova impacted a dry tropical forest of the Mexican Pacific coast in 2011 [37]. These strong relationships between wood density and severity of damage observed in TRF and TEF are in contrast to the result reported for subtropical tree species, that is, trees with the highest wood densities of 0.83 g/cm had the lowest survival rate (i.e., 22%) and highest branch loss (i.e., 60%) in an urban forest [38]. This inconsistency may be due to the level of exposure of trees to windstorms in the urban area compared with the natural forest setting. The study of [39] also reported contrasting result, i.e., low-density wood exhibited high resistance to uprooting, which was attributed to less rigidity of trees and significant loss of larger branches. Larger DBH and longer stems of light-demanding pioneer species may be more susceptible to windstorm damage than slender stems of shade-tolerant or low light-demanding trees. To reach the canopy rapidly, pioneer species may invest more in diameter and height growth for short-term gains and competition, thereby investing less in wood tissues for mechanical support (and thus have low wood density and low cost-stems). In contrast, small and intermediate trees growing below the tallest and largest ones may require low light levels, resulting in reduced diameter growth and higher wood density. During windstorms, thinner and denser stems can, thus, reduce the damage by reducing the sail area during short gusts of wind and enabling them to bounce back after having been struck by falling branches [40]. Results of the previous study also stipulated that dense woods have a higher modulus of rupture, which allows wider crowns while maintaining mechanical stability compared with less dense woods [41]. Partly, this can explain the higher mortality rate of trees with > 60 cm DBH observed in most study cases in TRF, SUF, and TEF (Figure 5). Higher mortality rates were reported for pioneer tree species with low wood density in the Luquillo Experimental Forest (LEF) of Puerto Rico as a result of tropical cyclone Hugo [42].

#### 3.1.2. Species and Population-Level Effects

Depending on the severity of the damage at the individual trees, the more serious effect can be seen at the species and population levels through windstorm-induced tree mortality and changes in stem density, causing a conspicuous influence on population demography or structure (Figure 4). This is because the removal of individuals in the forest area affects the spatial distribution of tree species. Increased tree mortality is one of the most evident effects of windstorms on forests in the form of tree suppression and mechanical damage, which predominantly depends on wind intensity and frequency [43] and tree size and species [44]. Here the majority of the reviewed studies indicated that the most-damaged tree species (both hardwoods and softwoods) decreased in stem density regardless of tree size due to tree mortality. Removal of trees in the forest can alter seed dispersal as influenced by tree distance and localized gap, resulting in fluctuations in stem regeneration rate, time, and stem abundance in the understory. Effects of tree mortality on population structure vary depending on which age, diameter, or height class and species survived or removed in the stand. When most of the small trees were damaged and killed by windstorms, for example, one potential effect is to retain the population of upper age, which tends to be more vulnerable to other causes of death (e.g., lightning, drought, fire, and pathogen attacks) due to old age.

Windstorm disturbance often leads to sudden increases in litterfall. Other reviewed studies attributed the massive litterfall deposition to decreased species diversity and stem abundance, by inhibiting seed germination through chemical and physical effects, such as covering seeds and killing of seedlings that are already present in the forest floor [45]. In this review, 26% of the study cases in TRF reported higher mortality in seedlings to saplings compared with the other DBH classes (Figure 5). Majority of the studies ascribed it not only to resource availability, susceptibility to windstorm damage, tree community and forest ecosystem attributes but also to the damage by falling emergent trees or massive litter deposition, particularly in tropical forests (e.g., [46,47,48,49]). The effects of massive litter accumulation and climatic factors on seedling recruitment were also investigated in a tropical forest, and results showed that, under the canopy, the removal of litter layer increased seedling emergence [50].

Several studies have also suggested that young secondary tropical forest in Puerto Rico experienced lesser damage from a hurricane than that of old secondary forests due to the low and uniform canopy and smaller-DBH trees [45]. Higher mortality in seedlings and saplings was also attributed to torrential rainfall with strong winds, which can wash away and suppress young regeneration in the understory (e.g., [51,52]). This implies that the generalities regarding the trajectory of forest structural changes caused by catastrophic windstorms may be impossible to date due to many interacting factors and, if not too few studies exist, the results of available studies have opposing conclusions. In temperate forests, for instance, the degrees of structural change vary significantly depending on wind intensities, rainfall amount, site biophysical conditions, and tree species characteristics [43,53]. The LEF and the Harvard Forest in Central Massachusetts, which both occasionally subject to the same catastrophic windstorms, are interesting examples of how windstorm damage varies depending on many biotic and abiotic factors in tropical and temperate forests. The LEF experienced greater damage than that of Harvard Forest due to higher susceptibility to wind damage, and that susceptibility was attributed to early leaf senescence, higher topographic exposure, and year-round warmer temperatures and higher precipitation [54].

Moreover, decrease and/or increase in stem density may also alter population genetics (Figure 4) through some indirect processes, including the removal of genetically superior and/or inferior trees [18,55] and spread of fast-growing invasive species [56]. These two processes may determine the number and quality of seedlings that can be recruited into a given area. Removal or decrease in stem density of inferior trees (e.g., poor health due to pathogen infection, abiotic stresses) may lead to the replacement of these trees with fitter ones that are already present at a given site or by genotype migration [57]. A study of [58] reported positive correlations among population size, its fitness and within-population diversity.

#### 3.1.3. Community and Ecosystem-Level Effects

The effects of catastrophic windstorms at the lower level of biological organization have the potential to affect forest successional patterns, which may also influence the forest species composition and community structure dynamics (Figure 4). Most of the studies reviewed, particularly those conducted at the LEF, reported that plant community establishment in windstorm-disturbed site begins with significant crown damage of adult trees and creation of gaps, followed by early emergence of pioneer species and gradual replacement by shade-tolerant species [5,59,60]. Field and remote sensing data have also shown a positive and strong correlation among windstorm disturbance measures (e.g., gap characteristics and tree mortality) and a fraction of resprouters, floristic composition, and species diversity four years after the catastrophic windstorm in central Amazon forest [61]. Similarly, high stem recruitment in both new and older gaps was reported for a temperate forest due to increased light levels compared with areas under closed canopy [62]. The frequency of occurrence of many understory species also significantly increased in both pine and oak forests, but the frequency of disturbance specialist species was higher in the pine forest 14 years after catastrophic windthrow [63]. A similar observation was observed three years after cyclone Hudah struck Masoala forest in Madagascar, i.e., an increase in the frequency of woody pioneer species (e.g., *Trema orientalis* Linn. Blume) [64]. When large trees, which are often the mother trees, are removed, the offspring growing in their vicinity may experience a substantial increase in abundance during the first few years following windstorm disturbance based on gap-phase regeneration theory [65,66,67]. The high-light conditions created by fallen trees are considered a critical determinant of seedling establishment and community structure dynamics [68,69]. However, not all windstorm disturbances can cause significant variation in light availability between gaps and non-gaps forest understory due to varying frequency and intensity of windstorms [65]. This can be exemplified in the study of [70] in which the species richness and other diversity indicators have no significant difference before and after the disturbance of hurricane Beta, and was consistent with the study of Imbert and Portecop [71]. Based on the review of Xi et al. [43], we can say that such results may be due to the lower intensity of the hurricane (i.e., Category one) compared with the intensity reported in the other reviewed studies, with categories ranging from three to five. Moreover, species richness decreased in all secondary forests in Puerto Rico, especially shrub species, based on short-term (4–5 years) response data [46]. The author attributed the result to the limitation of pioneer species recruitment in young secondary forest. This is because the changes in species richness in such a forest depend more on the composition of the seedling bank, rather than immediate recruitment of pioneers following wind disturbance [45]. In a young secondary tropical forest, the loss of trees, particularly those under small size classes, may result in the loss of important species that cannot be compensated by the regeneration of early successional ones. However, decrease in stem density due to tree mortality can also stimulate germination of seeds from both external sources and soil seed banks [72], depending on the age, life-history characteristics, land-use history, physical environment, and storm’s timing, speed, and intensity [73]. A recent review, for instance, reported that the regeneration and survival of species after disturbance depend on a species’ life-history traits [74].

Severe defoliation, snapping off of branches, and uprooting also pose significant alterations and/or fluctuations in the carbon and nutrient fluxes due to increased litter inputs in the forest floor (Figure 4). The process begins at improved litter decomposition rate in windstorm-hit areas as a response to a substantial increase in gap size and conspicuous increase in understory light levels [43,75]. Several studies have shown that areas exposed to strong winds, intense rainfall, and high air temperature increased litter decomposition rate (e.g., [76,77]). Coarse woody debris after windstorm disturbance can also provide additional habitat and food for microorganisms, detritivores, and soil insect communities, thereby increasing the rate of litter decomposition. Thus, the massive litter production associated with the effects of windstorms may significantly improve C and nutrient pools in the soil, and nutrient resorption [78,79]. However, massive litterfall does not always result in improved nutrient fluxes as it can also, in some instances, cause inaccessibility and immobilization of nutrients [14,80,81]. This is because plant species in communities differ in wood and leaf traits; hence, they also differ in many aspects including responses to windstorms and rates of decay. Different communities of decomposers (e.g., bacteria) also have contrasting feeding preferences depending on litter quality [82]. Hurricane-induced depositions of litter in Costa Rica’s primary forests have shown a significant reduction in the richness of decomposers of certain genera while selecting only the dominant genera possibly better suited to decompose the litter material [83]. Further, decay rates of dead standing and downed trees depend on many interacting factors, such as log size, local climate, and whether logs are in contact with the ground or saturated [84].

#### 3.1.4. Landscape-Level Effects

Few studies reported the apparent effect of windstorms on atmospheric carbon dynamics through either biomass destruction or lost carbon sequestration capacity of forested landscapes (Figure 4). Generally, reviewed studies have indicated that severe storms can enhance, increase, or maintain sequestration capacity of forested landscapes via changes in site conditions that are relevant to tree growth (e.g., nutrient inputs, species composition, and rates of recruitments) [23,85,86,87]. This is because forest fragmentation and vegetation cover change caused by windstorms can affect the floristic and functional attributes of tree communities via edge effects, with possible effects on forest landscapes’ total biomass. In this review, a high percentage (30–50%) of the studies reported that trees with DBH larger than 60 cm exhibited high mortality rate and crown deterioration in all forest types (Figure 5). The loss of large trees, being the most vulnerable to wind disturbance and microclimatic changes at forest edges, can negatively affect carbon stock and forests’ sequestration capacity [88,89,90], despite the high proliferation rate of pioneer species at edges [91]. Because these pioneer species are usually short-lived and small-sized, they cannot compensate for the biomass loss of larger trees with a much faster decay rate compared with the time required to gain equivalent biomass from tree growth in tropical and subtropical forests [73,89,92]. In the temperate forest, a few studies have shown that edge creation substantially increased forest biomass due to novel microenvironment conditions near or at forest edges (e.g., [93,94,95]), although some studies have also reported contrasting results (e.g., [87,96]). In cyclonic-and hurricane-zones where major wind disturbance is more frequent, however, some tree communities may increase the abundance of disturbance-resistant and-resilient species which are more adapted to edge effects [86,97], suggesting that the effects on biomass and forest growth may be dependent on the prevailing species composition and species life-history traits.

Studies have estimated the forest biomass loss due to the decrease in NDVI, uprooting of trees, snagging and breaking of tree boles and branches, defoliation, and reduction in leaf area [67,98]. A recent study reported that NDVI in exotic monoculture plantations decreased by >25% and only 12% of the landscape was unaffected by the typhoon, indicating a significant reduction in the green biomass in the landscape [50]. Several similar studies also reported a sudden significant drop in NDVI values after windstorms in the tropical, subtropical and temperate zones (e.g., [99,100,101,102]). The authors attributed it to the absence of native tree seedlings, high fire susceptibility and frequency due to dead trees and poor understory vegetation, and poor resistance to mechanical damage of exotic trees. Field experiments across geographical zones have already shown moderate to strong and significant correlations among green biomass, NDVI values, and percent groundcover [103,104,105].

Moreover, most of the reviewed studies also explained that massive and unsalvaged debris (particularly those calorific logs) can also heighten fire risk and insect outbreak, potentially increasing carbon losses and greenhouse gas emissions at the landscape level. This suggests that carbon sequestration in a windstorm-disturbed landscape may also depend on whether the windstorm debris is burned, completely decomposed, or consumed by insects.

### 3.2. Where Are We Now and How Should Research Move Forward?

#### 3.2.1. Knowledge Gaps

In the present literature review, we identified the gaps in information on the effects of catastrophic windstorms on structure, composition and dynamics of tropical, subtropical, and temperate forests. Majority of the reviewed studies were conducted in TRF, intermediate in SUF, and the lowest in TEF (specifically the countries in the Western North Pacific (WNP)). Further, although TRF has the highest number of study cases, most of the tropical countries found in the WNP have a low number of published studies regarding typhoon effects on forests. This result is relevant to improving both the amount and quality of studies in tropical and temperate forests located in the WNP because nearly one-third of the world’s total tropical cyclone (TC) activity occurs in the region, being the most active basin on Earth [106]. Based on the forecast data from the Japan Meteorological Agency (JMA), the frequency of rapid intensification (RI) cyclone events in the WNP increased from 1987–2018 [107]. Recent investigation has also projected a significant increase in the proportion of very intense TCs in the WNP as influenced by global warming [108,109,110]. The case of the Philippines exemplifies such increases in cyclone intensity as depicted in the unusual characteristics of 2013 Typhoon Haiyan (or Yolanda as its local name). From 1978 through 2013, Haiyan was one of the intense and destructive typhoons to have ever struck, not only the Philippines, but also most of the countries in the WNP based on track data from the main weather agencies in the region [111,112]. Globally, Haiyan’s intensity was among the highest ever reported for TCs [113] and was also unusual among other comparable events during the past six decades in the WNP [114]. These reports further justify the need to scale up the level of research in the tropical and temperate forests found in the WNP to better assess and quantify the impacts of windstorms on forests as the intensity and frequency of windstorm increase with global warming.

In terms of the most-studied topics by forest type/zone, researchers in TRF dealt mostly on tree mortality/survival and regeneration/succession, mortality/survival and species composition for SUF, and stem density and gap dynamics for TEF. The result on the number of study cases of these topics, which seems to increase at a faster rate with time, may serve as big data sets for each forest type, implying the potential role of big data technology (BDT) in research works on forest ecosystem management. As a global concern, forest management involves massive and complex data sets which the traditional data processing application software cannot deal with. Thus, systematic processing and extracting of information from large data sets about these most-studied topics, especially those data obtained from long-term monitoring ecological stations, can be carried out using BDT in TRF, SUF, and TEF. This can enable effective review and validation of the national and international forest statistics, and multi-scale identification of problems, constraints, and capacity needs. In this manner, the national and international forest management efforts can be linked and implemented more effectively across forest type/zone, particularly those countries that lack spatial and temporal data sets.

In all forest types, studies at the molecular-cellular-individual level were less common compared with those of the other levels of biological organizations. This can yield equivocal evidence on tree vulnerability to windstorm disturbance and severity of the damage. In particular, studies on roots and fungi interactions (e.g., mycorrhizae), population genetics and cellular and physiological mechanisms of plants (e.g., non-structural carbohydrates, photoinhibition, and photosynthesis) were very rare. Such studies are important because they are relevant to understanding the synergistic effects of windstorms on forests. The molecular-cellular-individual level of studies may also contribute plausible insights into selection pressure and evolutionary processes of windstorm-resistant tree species in TRF, SUF, and TEF. For example, research on changes in genetic diversity of self-regenerating populations of Norway spruce following disturbance provided a basis for the formation of ecologically sound stands that are able to adapt to ever-changing climatic conditions, particularly those populations with high genetic diversity and random spatial genetic structure [55]. It has also long suggested that chronic wind may lead to genetic differences within a species and adaptation to wind among contrasting species due to induced acclimation and selective pressure [14]. Moreover, tree populations rely on many factors, including phenotypic plasticity and physiological adaptation or metabolic adjustments within the cell or tissues to survive in extant locations or new environmental conditions [57,115]. A simulation study has shown that storms increased photosynthesis rate by up to nine percent in years with more tropical cyclones [116]. Furthermore, high concentrations of carbohydrate reserves can be detected in the trunks or roots of trees after physical damage, such as natural disturbances [117,118]. Both morphological and physiological traits are relevant to plant adaptation to windstorms, but as to which traits are the most vulnerable remains obscured. The vulnerability may also vary depending on the forest geographical locations and windstorm characteristics due to differences in site and environmental conditions and plant traits. Hence, further research is needed at the molecular-cellular-individual level with consideration of the spatial context of study locations to scale up our understanding of the vulnerability of plants and forest ecosystems to windstorm disturbance.

At the higher level of biological organizations, what is less common are studies that are relevant to C and nutrient fluxes via litterfall, litter decomposition, belowground ecological processes (e.g., fine root production and soil seed bank), plant invasion, and tree health/vigor in almost all forest types. First, more comprehensive research is needed on C and nutrient fluxes, litterfall production, and decomposition because windstorm disturbance regime and resource availability tend to interact, creating conditions suitable for growth and fitness of the self-regenerating individuals. Litterfall nutrient flux can, thus, provide insight on substrate-specific effects, the temporal requirement of recovery, and trajectories of the structure, composition, and function of the forest. The frequency and intensity of cyclones can also increase with global climate change; therefore, information on litterfall dynamics is essential for better understanding of the effects of windstorms on forest nutrient cycling and productivity, particularly in areas prone to multiple, climatic disturbance events. Second, only a few studies have addressed questions relevant to interactions of belowground ecological processes and windstorms despite their potential implications for predicting differences in species and ecosystem responses. For example, studying fine root biomass, production, and mortality in different environments, can provide explanations of why and how species vary in their performance and rate of recovery following windstorm disturbance. A study showed that fine root biomass recovered and increased seven months after an intense hurricane [81], and an increase in fine root mass may represent a good strategy to enhance soil resource uptake and plant performance [119]. Considering the intra- and interspecific differences in tree performance, further investigation is needed on the effects of catastrophic windstorms on tree roots to elucidate the difference in coping strategies between aboveground and belowground components of plants across plant taxa and forest types. Another research area that is urgently needed is on the ability of windstorms to facilitate the spread of invasive plant species, which often respond positively to disturbed ecosystems. Windstorms usually occur near the end of the growing season or when seeds are already mature, thereby increasing the risk of plant invasion through wind dispersal. Although the spread of exotic organisms has long been linked to windstorms, there is a need to update the information because some of the highly cited publications are relatively old; hence, the results may no longer be applicable due to the influence of climate change on windstorm characteristics. Consideration of biological invasion on future research can further explain major adverse effects of windstorms on forest structure, species diversity, tree population genetics, and forest health, especially in the temperate zones. Lastly, research on forest health in the aftermath of windstorms must be increased to expand the knowledge regarding the effects of windstorms on tree mortality or forest dieback, decay, and reduction in tree growth parameters. This is because a number of questions remain unanswered in most of the reviewed studies; for example, whether the recorded tree mortality is caused by windstorm or tree disease and whether the reduction in tree growth and species diversity is caused predominantly by windstorm-induced mortality or poor health conditions of trees or by the combined effects.

#### 3.2.2. Gaps in Methodological Approaches

In this review, we also found gaps in methodological approaches used investigating the effects of windstorms on forests. Although most of the studies were conducted in permanent plots in three forest types, a very small proportion of studies used controlled study approach with short study duration (i.e., 2–6 years for TRF and 5–7 years for TEF), except for SUF (i.e., 4–14 years), for studies about tree mortality/survival, stem density, species composition/diversity, regeneration/succession, gap dynamics, biomass and productivity, and BA/DBH/height. There was also a conspicuous gap in research about the topics reviewed beyond 2005. Therefore, increasing the number of controlled study (manipulative experiments) with longer duration in TRF and TEF must be taken into consideration in future research. Unlike the observational approach, the long-term controlled study can provide fundamental information on before, after, and control conditions of the site, following a BAC design. This study approach observes the conventions of appropriate sample size, replication, and where appropriate, the use of untreated or reference sites to provide results on windstorm impacts with a higher level of certainty compared with the observational one. Formal quantitative long-term monitoring, using pre-identified indicators, can also be undertaken under a controlled study approach to adequately observe the successional or developmental states of the forest’s decline or recovery following windstorm disturbance. However, this type of study may involve expensive and exclusive instrumentation (i.e., able to withstand strong wind) and laborious fieldwork.

A higher proportion of studies have successfully continued long-term observational studies in some established permanent plots in Luquillo Experimental Forest in Puerto Rico for TRF, Fushan Experimental Forest in Taiwan for SUF, and Tomakomai Experimental Forest in Japan for TEF. Research development investments in these permanent plots should be further encouraged along with the other least-studied permanent plots, particularly in windstorm-hotspot countries that have a low number of study cases; for example, Vietnam, Philippines, China, Mexico, and Australia. More long-term monitoring studies in permanent plots are important because the damage caused by the previous windstorm disturbance affects tree and ecosystem resistance in the succeeding storms, thus, storms should not be treated as discrete and independent events. Cumulative effects must be monitored and accounted for regularly. Innovations may also be employed, such as a combination of controlled, observational, and theoretical approaches and the use of advanced methodological techniques including drone-LiDAR (Light Detection and Ranging) system, terrestrial laser scanning, and other time-series photography technologies with high spatial, spectral, and temporal resolution. These remote sensing methodologies can effectively secure an image of the site before and at intervals after the windstorm disturbance, tracking actual changes over time. Here although a number of landscape-level studies have used Landsat 5–8 (can acquire satellite imagery of the earth), most of them are observational studies and conducted in temporary plots. On future research, the use of remote sensing technologies combined with field-based controlled study in permanent monitoring plots can, thus, improve effect detection accuracy and impacts and risks assessments, particularly for accurately quantifying the spatial extent of a disturbed forest at a landscape level.

Lastly, we found that landscape-level research is also lacking. Since it is extremely strenuous to monitor complex interdependent effects of windstorms occurring at multiple spatio-temporal scales, combined remote sensing techniques and mathematical modeling can be used as a complementary approach to effectively quantify changes at a landscape level. These approaches can model and describe successional trajectories, biomass loss, forest structure development, and forest dynamics over time.

## 4. Methodology

### 4.1. Data Collection

A systematic literature review was conducted to examine, assess, and summarize the state of available information and understanding of a select topic and research questions from peer-reviewed journals published between 2000 and 2020. Specifically, we followed, with some modifications, the systematic review process used by Lopez–Marrero regarding ecological research on hurricanes and forests in Puerto Rico [120]. To the best of our knowledge, her work was the first and only article to report about the effects of windstorms on forests, with detailed information on the systematic literature search process. In this review, the search databases used were Science Direct and Scopus, which are the leading sources and databases for peer-reviewed scientific studies [121,122]. At least two databases should have to be searched in a systematic literature review according to the guidelines [123]. Similar to our review, the Science Direct and Scopus databases were also the only databases used to source articles for a systematic review of the global trend of forest ecosystem services valuation [124]. We also found that these two databases were frequently included in the list of databases used for systematic review articles that are relevant to forestry (e.g., [125,126]).

After some pilot search and refinement of search terms, we formulated the final search terms by considering the geographical zone/forest type and windstorm of interest as shown in Table 1. In the case of windstorms, some terms including storm and windthrow or wind-throw were also used to improve the online search. Boolean search strings (i.e., AND, OR, and -) were used to join the search terms or keywords and collect multiple related works of literature. All these Boolean operators were entered in all uppercase in the databases’ search engine. In this review, the use of AND Boolean operator was intended to search for all of the articles that include both the terms (e.g., “windstorm” AND “tropical forest”, “typhoon” AND “ temperate forest”). On the other hand, we used OR Boolean operator to search for the articles that used either of the two search terms (e.g., Windstorms OR “typhoon”, “tropical cyclone” OR “hurricane”) in the entire paper. Further, search terminologies had run with the required number of strings per the requirements and limitations of the database used (e.g., Science Direct database allows only UTF-8 characters). In both databases, we used the advanced search form by specifying the keywords or search terms, publication year range, and article type.

### 4.2. Article Selection and Appraisal

The journal articles were reviewed, screened, selected, and evaluated based on the Preferred Reporting Items for Systematic Reviews and Meta-Analyses (PRISMA) guidelines using strict inclusion or exclusion criteria (Figure 6). The application of such criteria enabled the review work to narrow down the search results to the most relevant research articles. In the first step, a preliminary identification of relevant papers was done in the databases through title-abstract-keyword search strategy, through which databases would search only these segments of the articles that contain the terms. Plus, at this stage we limited our search to only peer-reviewed journal articles published between 2000 to 2020 written in English and conducted in temperate, subtropical, and tropical zones/forests, excluding conference proceedings, editorials, reference works, research notes, short communications, letters, and books through the advance search form. The exclusion of such types of literature had no significant effect on the availability of published information and understanding about the topic reviewed. All the articles that fulfill the inclusion criteria were selected for further investigation and content appraisal using the title screening search approach. Here the duplicates of the articles, as well as those irrelevant to the formulated research questions (e.g., irrelevant geographical zone), were removed by importing the search results to Endnote library (v. x9.2). The articles published in the same year and/or journal with the same title and author were deleted at this stage. The remaining articles after this stage were exported to an excel file. For some cases, the title, author, year, and name of the journal were encoded in the excel file to manually remove the duplicates. The articles resulting from title screening stage were then validated using the abstract screening search strategy under the validation stage. All abstracts were read and those with either ambiguous or no clear mention of the results about the effects or impacts of windstorms on forests in tropical, subtropical and temperate zones were excluded. The next stage was the quality assessment by which the full-texts of the remaining articles were read, particularly the results and methodology sections. We ensured that the remaining articles have good results with clear methodological approaches. Lastly, the Science Direct and Scopus databases provide links for free to access full-text articles in PDF format, and in case not found, we searched in some research websites.

Although we intended to manage a comprehensive review, some studies may have inadvertently been removed due to the multidisciplinary nature of publication outlets for windstorm disturbance literature. Thus, a manual search was done in two ways; using the Google search engine (majority from Research Gate, and Google Scholar) and references listed in the literature cited of the included studies or backward reference searching (also known as chain searching). Manual search also underwent further scrutiny following the inclusion/exclusion criteria (Figure 6).

### 4.3. Data Categorization

The full-texts of the 161 included articles were read to summarize the data based on three categories in a matrix format (Appendix A). The first main category (i.e., geographical zones /forest functional types) has three subcategories; namely, temperate, subtropical, and tropical forests/zones. We used the Köppen climate classification system to categorize the geographical zones of each article by extracting the coordinates of the study sites mentioned in the paper. Thus, studies conducted between 0° and 23.5° (between the tropics), 23.5° and 40°, and 40° and 60° latitudes were categorized as tropical, subtropical, and temperate forests/zones, respectively. The second main category deals with the levels of biological organization which was subcategorized into (1) molecular-cellular-individual level, (2) species-population level, (3) community-ecosystem level, and (4) landscape level. All studies that focused on the effects of windstorms on, for example but not limited to, tree population genetics, tree genetic diversity, wood anatomical structure, and tree physiology were grouped into the first level of biological organization (i.e., molecular-cellular-individual level). The species-population level included studies that dealt on, but not limited to, species and population growth rates. The studies that focused on the effects of windstorms tree species composition and diversity, forest stand structure, primary production, biomass, succession, carbon sequestration, gap dynamics, natural regeneration, nutrient cycling, litterfall production, and carbon and nutrient flux were also subcategorized into the community-ecosystem level. All data about changes in, for example, but not limited to, habitat or landscape cover or type were subcategorized under landscape-level effects of windstorms.

In terms of methodological approach, two subcategories were created; namely, observational and controlled experiment. The first subcategory refers to studies that used observational experiments, where the researchers conducted the study in areas where the effects of windstorms are already occurring [120,127]. In this review, we defined the controlled experiments based on the definition in [127], that is, studies that were done in the field, laboratory, or greenhouse with manipulated or controlled variables to measure effects. For studies that used secondary data for simulating or modeling the windstorm effects, we evaluated how the data was obtained (i.e., whether observational or controlled).

### 4.4. Data Analysis and Presentation

Data have first been reorganized using the Pivot table function in Microsoft Excel spreadsheet and analyzed using descriptive statistics (i.e., median, range, counts, and percentages) in R Studio Statistical Package Software (version 1.2.50119, 2009–2019 RStudio Inc., Free Software Foundation Inc. Boston, MA, USA). In filling-out the matrix, we made sure that the spelling of each entry to the main categories and subcategories were correct and consistent throughout the spreadsheet to avoid double counting. In some special cases, we allowed double/multiple counting; for example, all three target geographical zones were studied in one article. Thereafter, percentages were determined in terms of the total number of the selected articles or relative to the number of articles that belong to each main and subcategory. The statistics of the categorized data were calculated and presented in the form of tables and figures.

### 4.5. Study Limitations

The present systematic review was limited to original research articles published in English in scholarly journals between 2000 and 2020. All the analyzed articles are at least indexed in the Web of Science Citation Index, and the regional or local journals were not considered in this study. Nevertheless, we believe that the data we used and how they were analyzed can already provide a good overview of the extent of studies on windstorm and its effects on the forest ecosystem, and can already give valid research prospects and recommendations for future studies.

## 5. Conclusions

Recurrent windstorm is a complex disturbance that results in serious forest damage in the form of massive defoliation, stem breakage, toppling, uprooting, tree mortality, and canopy disruption, which influence forest stand structure, composition, and dynamics. Our review further confirms that windstorm effects are largely dependent on tree individual, population growth, and ecosystem traits, and windstorms characteristics. Effects of windstorms at the higher level of biological organization (e.g., ecosystem and landscape) synergistically emanate from the effects at the molecular-cellular-individual levels. Thus, the impacts of windstorms on the tropical, subtropical, and temperate forests can be better understood if the pattern of impacts at the lower level of biological organization is also considered. In terms of research gap, many basic yet important forest ecological processes relevant to understanding windstorm effects lack attention from the scientific arena across geographical zones. These research gaps include mycorrhizal symbioses, population genetics, physiological mechanisms/adaptation, carbon and nutrient fluxes via litterfall, litter decomposition, belowground ecological processes, biological invasion, and tree/canopy health. Moreover, controlled studies are particularly lacking in all the reviewed geographical zones but are more evident in the temperate zone, suggesting an area targeted to have an increasing amount of future research. Despite the higher number of studies in TRF, we found that some windstorm-hotspot countries in tropical zones also lack published studies about windstorm effects on the forest. Future investments in these research areas with enhanced methodological approaches, including instrumentation, can help explain resource use strategies and resistance of tree species to windstorms, ecosystem recovery and productivity, and quantify overall damage to forests. We also need to integrate advanced remote sensing and mathematical approaches, particularly at the landscape level, to meet the ever-changing challenges as a consequence of climate change. Therefore, there is much to be investigated about the topic reviewed to improve our understanding and inform management decision-making with regard to developing sustainable forests amid climate change.

## Figures and Tables

**Figure 1 plants-09-01709-f001:**
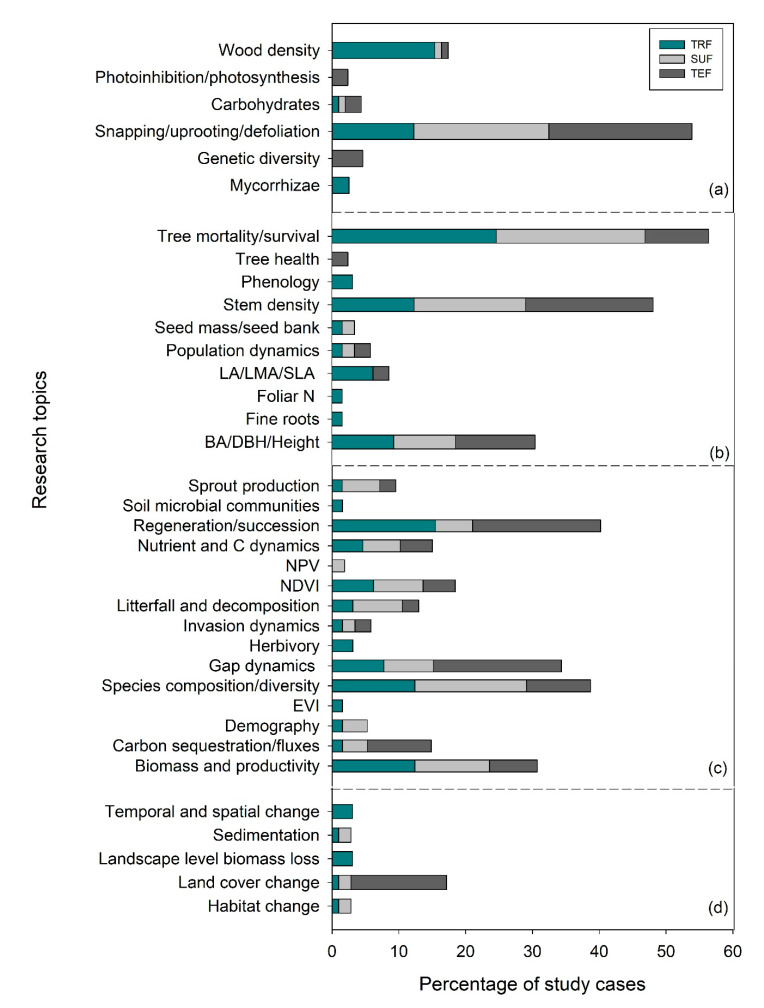
Percentage of study cases by research topics at (**a**) molecular-cellular-individual, (**b**) species-population, (**c**) community-ecosystem, and (**d**) landscape levels. Abbreviations: TRF—tropical forest, SUF—subtropical forest, TEF—temperate forest; LA—leaf area, LMA—leaf mas per unit area, SLA—specific leaf area, N—nitrogen, BA—basal area, DBH—diameter at breast height, NPV—nonphotosynthetic vegetation, NDVI—Normalized Difference Vegetation Index, and EVI—Enhanced Vegetation Index. Percentages were computed based on the total number of articles that belong to each main and subcategory.

**Figure 2 plants-09-01709-f002:**
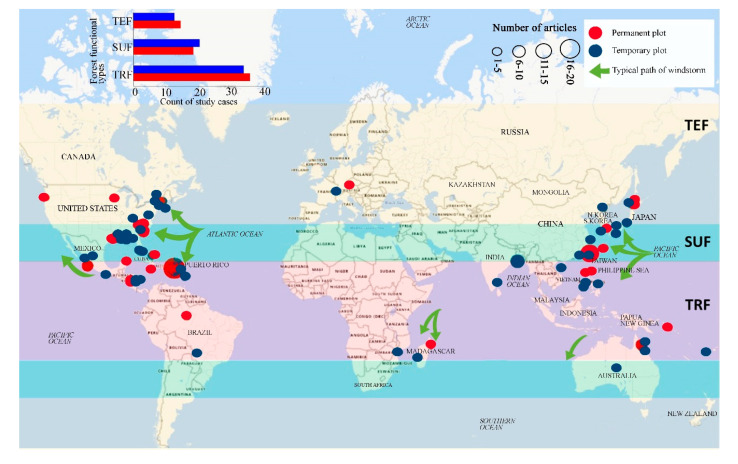
The number of study cases and global distribution of studies conducted in permanent/experimental and temporary plots in TRF, SUF, and TEF. The size of the dots is proportional to the number of studies for each type of plot.

**Figure 3 plants-09-01709-f003:**
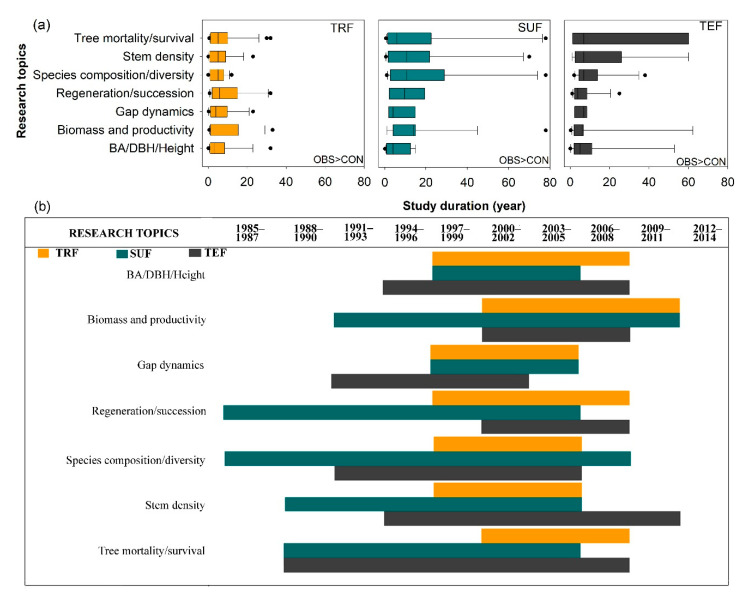
Study duration of the most-studied observational (OBS) and controlled (CON) research topics in TRF, SUF, and TEF: (**a**) descriptive statistics of the length of the study duration (year) and (**b**) time periods using the year data that appears most often in a set of data.

**Figure 4 plants-09-01709-f004:**
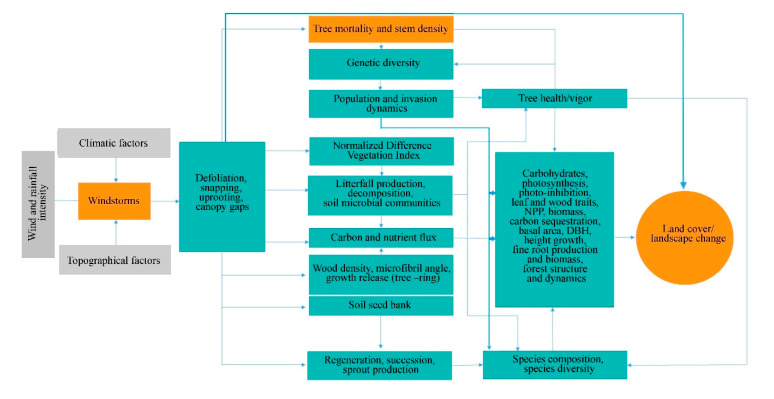
A conceptual diagram of the synergistic effects of windstorms on forest structure, composition, function, and dynamics.

**Figure 5 plants-09-01709-f005:**
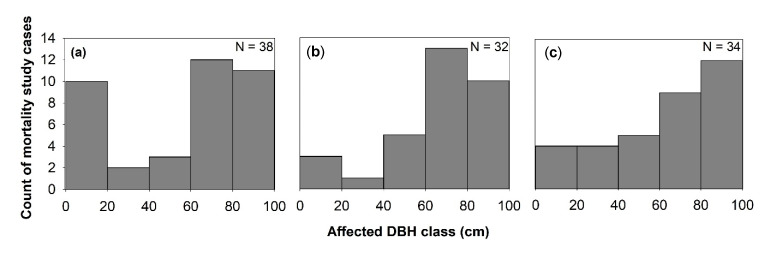
Windstorms effects on mortality across diameter at breast height (DBH) classes in (**a**) TRF, (**b**) SUF, and (**c**) TEF. N is the total number of articles that belong to each forest zone, and only studies that used % mortality were considered.

**Figure 6 plants-09-01709-f006:**
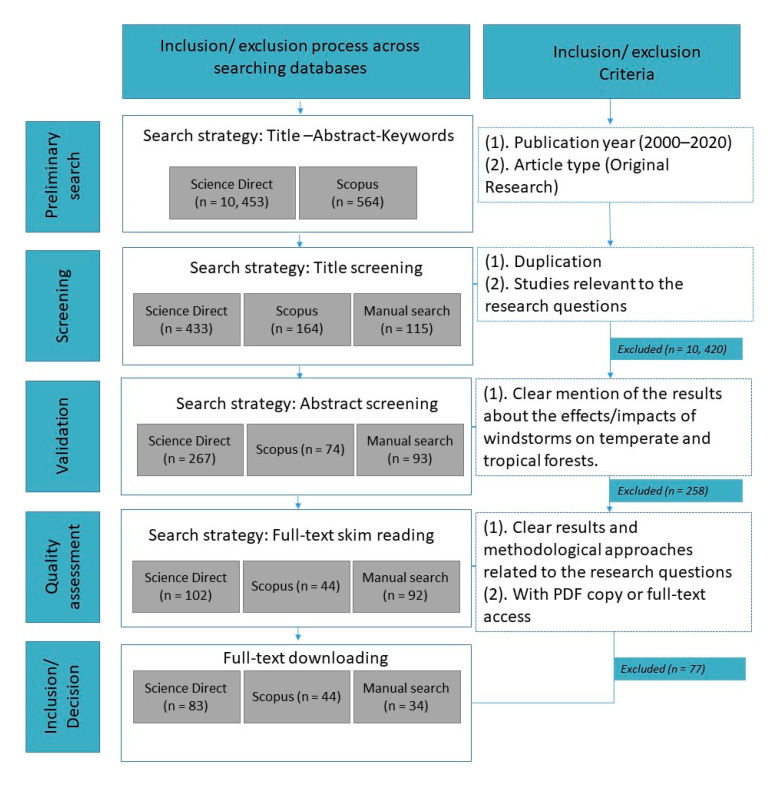
The flow diagram for database and manual search of publications used for the present systematic literature review.

**Table 1 plants-09-01709-t001:** The search terms and Boolean strings used in the systematic literature review using articles’ title, abstract, and keywords in Science Direct and Scopus databases.

Search Terms and Strings	No. of Articles		
	Science Direct	Scopus	Total
1. Windstorms OR “typhoon” OR “tropical cyclone” OR “hurricane” AND “tropical forest” OR “subtropical forest” OR “temperate forest”	5026	283	5309
2. Windstorms OR “typhoon” OR “tropical cyclone” OR “hurricane” AND “forest structure” OR “tree population” OR “tree genetic diversity” OR “tree diversity” OR “tree mortality”	2757	231	2988
3. Windstorms OR “typhoon” OR “tropical cyclone” OR “hurricane” AND “tree biomass” OR “forest carbon sequestration” OR “forest nutrient cycling” OR “tree regeneration” OR “tree species composition”	2636	49	2685
4. Windstorms OR “typhoon” OR “tropical cyclone” OR “hurricane” AND “gap dynamics” OR “tree demography” “tree primary productivity” OR “tree seedling growth” OR “seedling germination” OR “forest ecosystem structure”	34	1	35

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
