# Peer review of "Research Trends and Methodological Approaches of the Impacts of Windstorms on Forests in Tropical, Subtropical, and Temperate Zones: Where Are We Now and How Should Research Move Forward?"

_plants, 2020, doi:10.3390/plants9121709_

Round 1
Reviewer 1 Report
The paper addresses an important research question and is nicely written. There are some issues that need to address before consideration for publication.
- Abstract:
- Please delete “Windstorm disturbance does influence forest structure, species composition, function, and dynamics, and the magnitude of its effects at the ecosystem and landscape levels depend on the effects at the lower level of biological organizations (i.e., molecular-cellular-individual levels)”. This does not add any value and it is not your findings too;
- Please discuss key differences in findings between the tropical (TRF), subtropical (SUF), and temperate (TEF) forests/zones.
- While writing the abstract, please remember your three objectives. Some results from each objective should be documented in abstract.
- Introduction
- Nicely written
- Review methods
- Please discuss more on how you searched literatures. From which sources (e.g Web of Science, Scopus, Google Scholar, JSTOR and Science Direct) and why? Please follow these articles.
Paudyal, B.H., Maraseni, T. N., Cockfield, G. (2018) Evolutionary dynamics of selective logging in the tropics: A systematic review of impact studies and their effectiveness in sustainable forest management, Forest Ecology and Management 430, 166–175
Acharya, R., Maraseni, T.N., Cockfield, G. (2019) Global trend of forest ecosystem services valuation–An analysis of publications, Ecosystem Services, 39, 100979
- Discussions
- Discussion of the key differences of impacts of windstorms on tropical, subtropical, and temperate forests is poor. These differences should lead to guide separate (specific) research implications and recommendations each of these forest types (this means, you need to refine both “Where are we now and how should research move forward?” and Conclusion sections)
- Conclusion
- Nicely written
Author Response
Dear Editors and Reviewers
We are very thankful to the editors and reviewers for the valuable evaluation on the manuscript (plants-1010058), “Research trends and methodological approaches of the impacts of windstorms on forests in tropical, subtropical, and temperate zones: Where are we now and how should research move forward?”. We have tried to address all the comments and suggestions in a proper way and believe that the scientific rigor of the paper has improved further.
We would be happy to make further corrections if necessary and look forward to hearing from you soon.
Sincerely,
Byung Bae Park
Associate Professor
Department of Environment and Forest Resources
College of Agriculture and Life Sciences
Chungnam National University
99 Daehak-ro, Yuseong-gu, Daejeon 34134, Republic of Korea
Tel: +82-42-821-5747
Fax: +82-42-825-7850
Email: bbpark@cnu.ac.kr
On behalf of all authors.

Reviewer 2 Report
This manuscript is nearly ready to publish. Some moderate changes are needed, however. The largest issue I found is that the Methodology should precede the Results. See attached PDF for suggestions.

Author Response

(The authors gave the same response as above.)

Reviewer 3 Report
The authors present an interesting review of the impacts of windstorms on forests at the various integration levels and identify gaps in the knowledge. While the work is really full of information, it suffers however of insufficient or inappropriate organization. I am convinced that reorganization could produce an excellent paper.
The authors have decided to discuss the points around the levels of biological organization while their work begin by detailling the types of damages induced by windstorms. I think the paper would benefit of a structure organized around the damage types because these ones indeed mainly concern different levels of biological organization. For clarity also, the text should be divided in more subsections.
Some specific comments are given in the uploaded file.

Author Response

(The authors gave the same response as above.)

Round 2
Reviewer 1 Report
Nicely revised. One of the best articles I ever reviewed from MDPI journals.
Author Response
Thank you very much for your comments!

Reviewer 3 Report
I found this new version of the manuscript really improved, particularly, the structure selected by the authors now clearly appears and makes the text much more convenient to read. While I am not particularly qualified for English, I feel that it should be at least edited for minor changes for instance with definite articles in several places. See also L380 ‘most’ is missing at the beginning of the line.
The legend of Figure 3 should be corrected and improved. Suggestion:
Figure 3. Study duration of the most-studied observational (OBS) and controlled (CON) research topics in tropical (TRF), subtropical (SUF), and temperate forests (TEF): (a) descriptive statistics of the length of study duration (year), (b) time periods using the year data that appears most often in a set of data.
Figure 4. There are 2 red lines but it is not explained why in the legend.
